# CD44v6 High Membranous Expression Is a Predictive Marker of Therapy Response in Gastric Cancer Patients

**DOI:** 10.3390/biomedicines9091249

**Published:** 2021-09-18

**Authors:** Gabriela M Almeida, Carla Pereira, Ji-Hyeon Park, Carolina Lemos, Sofia Campelos, Irene Gullo, Diana Martins, Gilza Gonçalves, Dina Leitão, João Luís Neto, Ana André, Clara Borges, Daniela Almeida, Hyuk-Joon Lee, Seong-Ho Kong, Woo Ho Kim, Fátima Carneiro, Raquel Almeida, Han-Kwang Yang, Carla Oliveira

**Affiliations:** 1i3S—Instituto de Investigação e Inovação em Saúde, Universidade do Porto, 4200-135 Porto, Portugal; galmeida@ipatimup.pt (G.M.A.); clrc.pereira8@gmail.com (C.P.); clclemos@ibmc.up.pt (C.L.); Sofia.campelos@gmail.com (S.C.); irene.gullo12@gmail.com (I.G.); dianam@ipatimup.pt (D.M.); aandre@ipatimup.pt (A.A.); fcarneiro@ipatimup.pt (F.C.); ralmeida@ipatimup.pt (R.A.); 2Ipatimup—Institute of Molecular Pathology and Immunology of the University of Porto, 4200-135 Porto, Portugal; gilzasofia@gmail.com; 3Faculty of Medicine, University of Porto, 4200-319 Porto, Portugal; dinaraquel@med.up.pt; 4Doctoral Programme in Biomedicine, Faculty of Medicine of the University of Porto, 4200-319 Porto, Portugal; 5Department of Surgery, Seoul National University Hospital, Seoul 03080, Korea; pjhaaa1220@gmail.com (J.-H.P.); appe98@snu.ac.kr (H.-J.L.); wisehearted@gmail.com (S.-H.K.); hkyang@snu.ac.kr (H.-K.Y.); 6UnIGENe, IBMC—Institute for Molecular and Cell Biology, 4200-135 Porto, Portugal; 7ICBAS—Instituto Ciências Biomédicas Abel Salazar, Universidade do Porto, 4050-313 Porto, Portugal; 8Department of Pathology, Ipatimup Diagnostics, Institute of Molecular Pathology and Immunology, University of Porto, 4200-135 Porto, Portugal; 9Department of Pathology, Centro Hospitalar Universitário de São João, 4200-319 Porto, Portugal; 10Department of Biomedical Laboratory Sciences, ESTeSC—Coimbra Health School, Polytechnic Institute of Coimbra, 3046-854 Coimbra, Portugal; 11Instituto de Medicina Molecular João Lobo Antunes, Faculdade de Medicina, Universidade de Lisboa, 1649-028 Lisbon, Portugal; joaoneto@medicina.ulisboa.pt; 12Medical Oncology Department, Centro Hospitalar Universitário de São João, 4200-319 Porto, Portugal; claramlborges@gmail.com (C.B.); daniela.psa@sapo.pt (D.A.); 13Cancer Research Institute, Seoul National University, Seoul 03080, Korea; 14Department of Surgery, Seoul National University College of Medicine, Seoul 03080, Korea; 15Department of Pathology, Seoul National University College of Medicine, Seoul 03080, Korea; woohokim@snu.ac.kr; 16Faculty of Sciences, University of Porto, 4169-007 Porto, Portugal

**Keywords:** stomach neoplasms, genetic heterogeneity, biomarkers, drug therapy, conventional chemotherapy, gastric surgery

## Abstract

In gastric cancer (GC), biomarkers that define prognosis and predict treatment response remain scarce. We hypothesized that the extent of CD44v6 membranous tumor expression could predict prognosis and therapy response in GC patients. Two GC surgical cohorts, from Portugal and South Korea (*n* = 964), were characterized for the extension of CD44v6 membranous immuno-expression, clinicopathological features, patient survival, and therapy response. The value of CD44v6 expression in predicting response to treatment and its impact on prognosis was determined. High CD44v6 expression was associated with invasive features (perineural invasion and depth of invasion) in both cohorts and with worse survival in the Portuguese GC cohort (HR 1.461; 95% confidence interval 1.002–2.131). Patients with high CD44v6 tumor expression benefited from conventional chemotherapy in addition to surgery (*p* < 0.05), particularly those with heterogeneous CD44v6-positive and -negative populations (CD44v6_3+) (*p* < 0.007 and *p* < 0.009). Our study is the first to identify CD44v6 high membranous expression as a potential predictive marker of response to conventional treatment, but it does not clarify CD44v6 prognostic value in GC. Importantly, our data support selection of GC patients with high CD44v6-expressing tumors for conventional chemotherapy in addition to surgery. These findings will allow better stratification of GC patients for treatment, potentially improving their overall survival.

## 1. Introduction

Gastric cancer (GC) is the fourth most deadly cancer worldwide, with >750,000 deaths estimated to occur every year [1]. In the Western world, 70% of GC patients present with locally advanced and/or unresectable disease, for whom conventional chemotherapy is the treatment of choice, with a median overall survival (OS) of ~1 year [2]. In contrast, GC is mostly detected at early stage in East Asian countries that have implemented GC screening programs [3].

Despite improvements, targeted therapies have proved disappointing in GC [4], and those approved (trastuzumab against HER2, ramucirumab against VEGFR2, and pembrolizumab against PDL1) show yet limited OS improvement [4,5,6]. Despite the potential of immunotherapy in GC treatment, consolidated knowledge is only available for trastuzumab, as a predictive marker of therapy response. Patients with HER2 high-expressing tumors present longer OS in response to trastuzumab and chemotherapy than those with HER2 low-expressing tumors [5]. This is further reinforced by the fact that patients harboring tumors with homogeneous HER2 expression respond better to trastuzumab than patients displaying HER2 heterogeneity [7].

MAPK-related activating alterations are frequent in colorectal cancer, particularly KRAS mutations, and are used as markers of non-response to therapy with EGFR-tyrosine kinase inhibitors, bringing benefit for these patients [8]. These alterations are rare in GC, and apart from HER2 overexpression, few markers of response to systemic therapy have been established [4,5,6]. Therefore, it is crucial to identify biomarkers that can better define prognosis, but mainly it is crucial to identify biomarkers that can better predict which patients are more likely to benefit from a given treatment regimen.

The human CD44 gene (NG_008937) encodes a polymorphic group of transmembrane glycoproteins generated by alternative splicing. The standard CD44 isoform (CD44s) includes only the constitutive exons, while the variant isoforms (CD44v) contain one or more variable exons (in addition to the constitutive ones) [9]. We have shown that CD44s is widely expressed in both normal and diseased gastric epithelial cells, while CD44v6-containing isoforms are de novo expressed in stomach premalignant lesions and in ~70% of all GCs [9]. Aberrant expression of CD44v isoforms has also been associated with several cancer-related features, such as invasion and metastases [10,11], as well as therapy response in gastrointestinal cancer cells [12,13,14,15]. An association between increased CD44-overall or CD44v6-specific immuno-expression in tumors and worse OS in GC patients has previously been reported [16,17,18]; however, some of the studies had low patient numbers and failed to demonstrate prognostic value independent of TNM staging. Also unknown is how patients bearing high CD44v6-expressing gastric tumors respond to conventional therapy regimens. The involvement of CD44v6 in all these cancer-associated processes suggests it is worth further studying CD44v6 as a potential biomarker of GC patient outcome and/or therapy response. 

Herein, we aimed to clarify the role of CD44v6 in GC by using two large GC cohorts from different world regions to investigate the relationships between CD44v6 expression and clinicopathological features, patient OS, and therapy response.

## 2. Materials and Methods

For details, please see Appendix A.

### 2.1. Patient Samples, Data Collection, and Tissue Microarray Preparation

Two GC patient cohorts were analyzed. The cohort from Centro Hospitalar Universitário de São João (CHUSJ), in Portugal, comprised 326 tumor samples from GC patients surgically treated between January 2008 and December 2014, stored at the Tumor Biobank of CHUSJ/Ipatimup [19]. The cohort from Seoul National University Hospital (SNUH), in South Korea, comprised 638 tumor samples from GC patients surgically treated between January 2010 and December 2011. Tissue microarrays (TMA) were prepared from formalin-fixed paraffin-embedded (FFPE) tumor material using an Arraymold Kit A (IHC World, Woodstock, Maryland, USA) [20] or a trephine apparatus. A total of 121/326 cases from CHUSJ and 272/638 from SNUH received platinum- and/or fluoropyrimidine-based chemotherapy in addition to surgery, mostly in an adjuvant setting (Appendix A). Clinicopathological, treatment, and survival data were collected from all patients. Studies were approved by the Ethics Committees of CHUSJ (CES 122/15 and CES 117/18) and SNUH (IRB: H1706-105-860), and informed patient consent was obtained from all patients. Tumor staging (pTNM—Pathological Tumor-Node-Metastasis) is reported according to the seventh edition of the American Joint Committee on Cancer classification system. This study was REMARK compliant.

### 2.2. CD44v6 Immunohistochemistry of GC Samples

Immunohistochemistry (IHC) staining for CD44v6 was performed in 3 µm TMA sections, using a mouse monoclonal antibody (clone MA54, 1:400; Invitrogen, Carlsbad, CA, USA). The assay was carried out on an automated Ventana BenchMark ULTRAStaining System, using the OptiView DAB IHC Detection Kit (both from Roche/Ventana Medical Systems, Tucson, Arizona, USA). The percentage of tumor cells displaying membranous expression of CD44v6 was assessed, and cases were classified as “CD44v6_0”—no staining at the cell membrane; “CD44v6_1+”—membranous staining in up to 10% of tumor cells; “CD44v6_2+”—membranous staining in between 11% and 50% of tumor cells; “CD44v6_3+”—membranous staining in between 51% and 75% of tumor cells; and CD44v6_4+—membranous staining in over 75% of tumor cells (Figure 1A).

### 2.3. Statistical Analysis

Categorical variables were analyzed using chi-square or Fisher’s exact test and continuous variables with Student’s *t*-test or one-way ANOVA (with Tukey’s post hoc test). Kaplan–Meier estimates of overall survival (OS) were obtained between groups. Multivariate Cox regression analysis of OS was performed and the hazard ratio (HR) and 95% confidence interval (CI) estimated to determine factors independently associated with OS. A *p*-value < 0.05 was considered significant. Analyses were performed using IBM SPSS Statistics versions 26 and 27 for Windows (IBM Corp, Armonk, NY, USA). This study was TRIPOD compliant.

## 3. Results

### 3.1. Analysis of De Novo CD44v6 Membranous Expression and Its Relationship with GC Patients’ Survival

Clinicopathological features, treatment regimen, and OS from two GC surgical series were collected from CHUSJ in Portugal (*n* = 326) and SNUH in South Korea (*n* = 638) (Appendix A). Both series had a good representation of all disease stages and, as expected, patient OS significantly worsened with increasing pTNM staging (Appendix A and Appendix A). Notably, GC patients from the SNUH cohort generally presented higher median OS or 5 year-OS rates than patients from CHUSJ (Appendix A). Additionally, as expected, patients who were treated with conventional chemotherapy in addition to surgery had extended OS, when compared with patients treated with surgery alone (*p* < 0.05) (Appendix A). To elucidate the role of CD44v6 in GC, we created a categorization algorithm for the extent of CD44v6 membranous expression in tumors that can be easily replicated. We classified tumors into four sub-categories, from absent (CD44v6_0) to very high (CD44v6_4+), as shown in Figure 1A (See Materials and Methods for details on classification).

We analyzed clinicopathological features according to the extent of CD44v6 expression in tumors from the CHUSJ cohort. We found that CD44v6_4+ tumors more often occurred in females (*p* = 0.005), presented increased depth of invasion (*p* = 0.010), and most often presented vascular (*p* = 0.005) and perineural invasion (*p* = 0.034) and that CD44v6_3+ and _4+ tumors have a higher frequency of tumor cells in the surgical margins (*p* = 0.007) (Table 1).

As tumors with 50% or more cells expressing CD44v6 (CD44v6_3+ and, particularly, _4+) seem more invasive, it is likely that they present worse OS than other subgroups. We addressed this and found that patients bearing tumors with lower CD44v6 expression (1+ and 2+) present improved OS compared with those who have higher (3+ and 4+) or lack CD44v6 expression (0) (Figure 1B). In fact, there is an overlap between the survival curves representing CD44v6_1+ and CD44v6_2+ and between the survival curves of CD44v6_3+ and CD44v6_4+ patients (and no significant changes between them: *p* = 0.604 and *p* = 0.663, respectively; Figure 1B and Appendix A). Therefore, it seemed reasonable to group patients with CD44v6-positive tumors into two CD44v6 expression categories (low and high), according to prognosis, and use expression in 50% of the tumor cells as the cutoff: CD44v6 < 50% or CD44v6_Low (1+ and 2+ cases) and CD44v6 ≥ 50% or CD44v6_High (3+ and 4+ cases) (Figure 1A). As highlighted in Figure 1C (and Appendix A), GC patients with CD44v6_Low tumors presented significantly better OS than patients with higher CD44v6 expression (median OS ~68 months vs. 26 months, *p* = 0.002) or lacking CD44v6 expression (median OS ~68 months vs. 30 months, *p* = 0.004). Clinicopathological associations, described in Table 1, are maintained when grouping patients into these three CD44v6 categories—namely, the association between CD44v6_High tumors with invasive features (perineural invasion, vascular invasion, positive surgical margins and depth of invasion); the only exception being association with gender (Appendix A). Additionally, multivariate analysis shows that CD44v6_High is a borderline independent factor of poor prognosis in this patient cohort (*p* = 0.049) (Figure 1D). Additionally, pTNM stage (high) (*p* < 0.0001), mean age at onset (higher) (*p* < 0.0001), and surgical margins (positive) (*p* = 0.01) were identified as strong independent poor prognosis factors.

Analysis of clinicopathological features of the SNUH GC patient cohort according to CD44v6 tumor expression showed: a higher proportion of women with CD44v6_3+ tumors (*p* = 0.009); increased depth of invasion in CD44v6_0 and CD44v6_4+ tumors (*p* = 0.0001); increased presence of lymphatic permeation (*p* = 0.001) and lymph node metastases (*p* = 0.008) in CD44v6_4+ tumors; and an enrichment of pTNM stage III in CD44v6_0 and CD44v6_4+ tumors (*p* = 0.001) (Table 2).

Thus, both CD44v6_0 and _4+ tumors present increased depth of invasion and higher pTNM stages (particularly III), while increased lymph node metastasis and lymphatic permeation is specific to CD44v6_4+ tumors. These associations seem particular to the most homogeneous subcategories (0 to 4+), as some are no longer identified when CD44v6 status is grouped into absent, low, and high (Appendix A).

When analyzing the prognostic value of CD44v6 in these GC patients, we found no differences in patient survival between the different CD44v6 subgroups (CD44v6_0 to CD44v6_4+), or when they were grouped in CD44v6_Low to CD44v6_High (*p* > 0.05) (Appendix A). In the SNUH GC patient cohort, multivariate analysis results identified high pTNM stage (*p* < 0.0001), higher mean age at onset (*p* < 0.0001), having diffuse gastric cancer (according to Laurén classification) (*p* = 0.017), presence of lymphatic permeation (*p* = 0.009), and presence of perineural invasion (*p* = 0.006) as independent factors of poor prognosis, but not CD44v6 expression (Appendix A).

In summary, we identified a clear cutoff for defining overexpression of CD44v6 in GC (> 50% tumor cells overexpressing membranous CD44v6), which was a frequent feature in GC in both cohorts analyzed (27% for CHUSJ and 35% for SNUH). We demonstrated prognostic value for the Portuguese but not for the Korean cohort, likely due to the better OS of the latter, even when comparing the same pTNM stage.

### 3.2. Analysis of CD44v6 Expression in Tumor Cells and Its Relationship with Therapy Response in GC Patients

We wanted to assess whether GC patients respond differently to therapy, depending on the CD44v6 expression status. In the CHUSJ cohort, we observed that CD44v6-Low patients had better median OS regardless of whether they received chemotherapy in addition to surgery (Figure 2A,B), which was not the case in the SNUH cohort (Figure 2C,D). Indeed, when looking only at the CHUSJ patients who underwent surgical treatment alone (without additional chemotherapy), CD44v6_Low patients had a median OS of 94 months compared with 33 months in CD44v6_Absent patients (*p* = 0.027) and with 17 months in the CD44v6_High patients (*p* = 0.0002) (Figure 2A and Appendix A). When looking at the CHUSJ patients who received both surgery and conventional chemotherapy as part of their therapeutic strategy, CD44v6_Low patients still presented better OS compared with both CD44v6_Absent patients (median OS of 43 vs. 26 months, respectively; *p* = 0.025) and CD44v6_High patients (median OS of 43 vs. 36 months, respectively), although the latter did not reach statistical significance (*p* > 0.05) (Figure 2B and Appendix A).

We then compared the OS of patients receiving surgery alone with that of patients receiving chemotherapy in addition to surgery. In these analyses, only pTNM stage II and III patients were included since >95% of stage I patients herein analyzed remained chemotherapy untreated and would introduce bias. Following the same rationale, stage IV patients are commonly treated with palliative chemotherapy without curative intent and were also excluded. When directly comparing the OS of CHUSJ patients who received surgery alone, with that of patients who received chemotherapy in addition to surgery, per CD44v6 subgroup, we can observe that the added benefit of chemotherapy in improving patient survival was only significant when CD44v6 was expressed in over 50% of the tumor cells (Figure 3A–C). Indeed, administration of chemotherapy (in addition to surgery) to CD44v6_High patients resulted in a 3.3-fold increase in their median OS, from 12 to 39 months (*p* = 0.045) (Figure 3C). Although not as striking, the median OS of patients with CD44v6_Absent tumors almost doubled (from 20 to 40 months) when treated with conventional chemotherapy in addition to surgery (*p* = NS; Figure 3A). In contrast, when patients’ tumors presented CD44v6 expression in less than 50% of their cells (CD44v6_Low), there was no difference between the median OS of patients treated with chemotherapy in addition to surgery or with surgery alone (Figure 3B).

When analyzing the SNUH cohort, we observed the same trend (Figure 3D–F). Indeed, only administration of chemotherapy to CD44v6_High patients resulted in a significant increase in their median OS, from 81 months when patients are treated with surgery to over 120 months when patients are treated with chemotherapy in addition to surgery (*p* = 0.028) (Figure 3F). Similar to what was observed in the CHUSJ cohort, patients whose tumors lacked CD44v6 expression tended to benefit from receiving chemotherapy in addition to surgery (median OS of 55 months to over 120 months; *p* = NS; Figure 3D), and OS of patients with CD44v6_Low tumors did not improve, regardless of chemotherapy (Figure 3E).

Interestingly, within the CD44v6_High cases, this benefit was mainly associated with patients with CD44v6_3+ tumors in both cohorts (*p* = 0.007 for CHUSJ; *p* = 0.009 for SNUH) (Figure 4A–D). In contrast, when analyzing CD44v6_1+ and _2+ groups independently, no significant OS improvement was seen with “chemotherapy and surgery” vs. surgery alone (Appendix A).

Overall, data from both cohorts consistently denote that CD44v6_High patients (namely CD44v6_3+) particularly benefit from chemotherapy in addition to surgery, as opposed to patients from all other CD44v6_subgroups (Figure 4E,F).

## 4. Discussion

Herein, we established a protocol to evaluate the CD44v6 marker in GC according to the percentage of tumor cells expressing membranous CD44v6. This study demonstrated that CD44v6 was a predictive marker of response to conventional chemotherapy in two independent cohorts, and indicated it as a prognosis marker in a single cohort. 

We found that 80–90% of gastric tumors expressed membranous CD44v6, which was particularly high in ~27–35% of cases in both cohorts. Moreover, specifically in the Portuguese cohort, CD44v6_High patients presented significantly worse median OS compared with CD44v6_Low patients, independent of pTNM staging. This is corroborated by previously published meta-analyses in GC patients [16,17,18,19,20,21] and in other cancer types, such as lung, colorectal, and breast [22,23]. Additionally, two reports associated CD44v6 expression with worse OS and higher TNM staging [17,18], leaving reasonable doubt whether this is an independent factor of poor prognosis in GC. In addition, GC cohorts used to assess CD44v6 expression are generally small, which can compromise statistical analyses. Our study of larger GC independent cohorts supports CD44v6 as a poor prognosis marker in the Portuguese cohort, independent of pTNM staging, lymphovascular permeation, and perineural invasion, but not in the South Korean cohort. This discrepancy is likely unrelated to technical issues. In our study, IHC analyses and their interpretation were performed under the same technical and experimental conditions, although samples and TMAs were processed in different institutions. Moreover, tumor samples were collected from contemporary GC cohorts who underwent similar conventional chemotherapy regimens and included patients from all pTNM stages. Thus, the reason why CD44v6 was a marker of poor prognosis in one cohort and not in the other may be attributable to intrinsic differences between the two populations.

Independent of the controversy around the prognostic value of CD44v6 in GC, CD44v6_High was recurrently associated with tumor invasive properties in both cohorts, likely highlighting a behavior thought to be triggered by CD44v6 itself [10,11].

Our most important finding shows that CD44v6_High tumor expression predicts better response for patients treated with conventional chemotherapy in addition to surgery. This was particularly relevant for patients harboring CD44v6_3+ tumors and was validated in both GC patient cohorts. To the best of our knowledge, this is the first report demonstrating CD44v6 as a useful marker for predicting therapy response in GC, similar to what was shown in colorectal cancer, where patients with moderate or strong CD44v6 tumor expression responded better to irinotecan-based chemotherapy [24]. This is an impactful finding for a non-neglectable proportion of GC patients (14–21% of CD44v6_3+ cases) from either cohort, since up until now, only HER2 overexpression and microsatellite instability were widely accepted as therapy response markers in GC [4,8].

TNM staging has long been the most important tool for assessing prognosis in GC. Our data show that evaluating CD44v6 by IHC may provide oncologists with additional and important information for stratifying GC patients for treatment. Namely, if endoscopic tumor biopsies from GC patients show high CD44v6 expression, this information may be used to recommend surgery combined with chemotherapy, particularly for patients with extensive but heterogeneous CD44v6 expression (i.e., CD44v6_3+ tumors). Indeed, it appears that within gastric adenocarcinomas showing extensive CD44v6 expression, the most heterogeneous ones (i.e., CD44v6_3+) respond better to conventional chemotherapy when compared with the most homogeneous ones (i.e., CD44v6_4+). 

Our data also show that CD44v6 is a good marker for predicting longer OS in patients treated with curative intent (stages II and III) with chemotherapy in addition to surgery. Stage I GC patients from our cohorts rarely received chemotherapy in addition to surgery, so they were not included in these comparative analyses. Nowadays, TNM stage IB patients from the Western world have an indication for receiving chemotherapy in addition to surgery. Based on our results obtained with stage II and III, we believe that stage I GC patients bearing CD44v6_3+ tumors are preferential candidates for chemotherapy in addition to surgery. This marker could also be considered as a predictor of better survival outcomes in stage IV fit patients treated with chemotherapy. In both cases, additional clinical studies are needed. 

From a patient’s perspective, it would be important to inform those with CD44v6_3+ tumors that their tumor molecular characteristics are likely predictive of extended survival if chemotherapy is added to surgery. This would help these patients to cope with disease-associated psychological and emotional challenges.

## 5. Conclusions

Our study is the first to identify CD44v6 high membranous expression as a potential predictive marker of response to conventional treatment, but it does not clarify CD44v6 prognostic value in GC. Importantly, our data support the selection of GC patients with high CD44v6-expressing tumors for conventional chemotherapy in addition to surgery. These findings will allow better stratification of GC patients for treatment, potentially improving their OS.

## Figures and Tables

**Figure 1 biomedicines-09-01249-f001:**
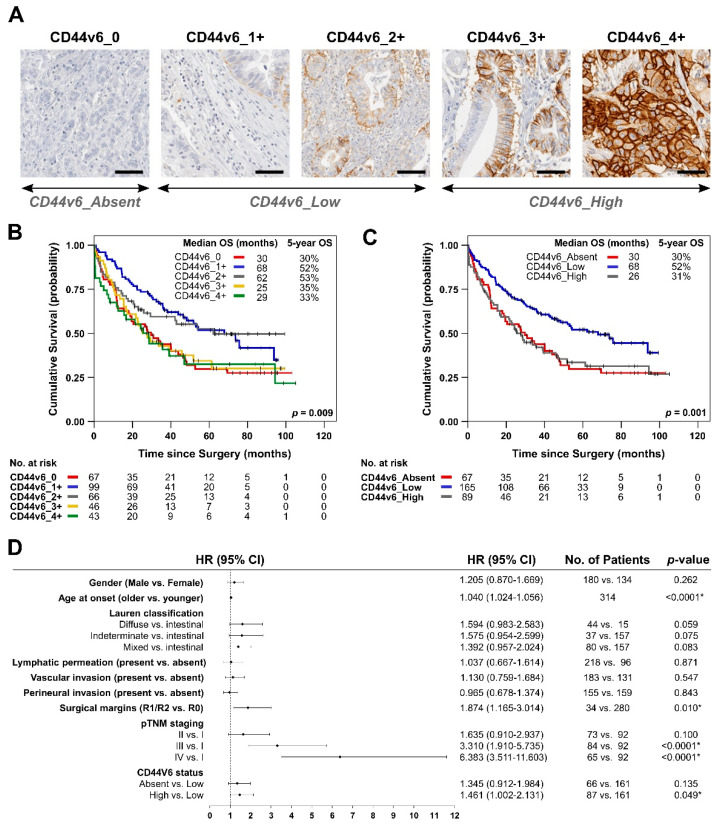
(**A**) Immunohistochemistry characterization of tumor samples for the presence and extent of CD44v6 expression. Images are representative of the five categories defined. Scale bar represents 50 µm; Kaplan–Meier estimates showing OS from CHUSJ GC patients, according to CD44v6 sub-categories (**B**) or according to CD44v6 status (absent, low, and high expression) (**C**). Median OS and 5-year OS are shown for each patient subgroup. *p*-values for pairwise comparisons by the Log-Rank (Mantel–Cox test) are shown in Appendix A, respectively; (**D**) Forest plot of the multivariate analysis. * highlights statistically significant differences.

**Figure 2 biomedicines-09-01249-f002:**
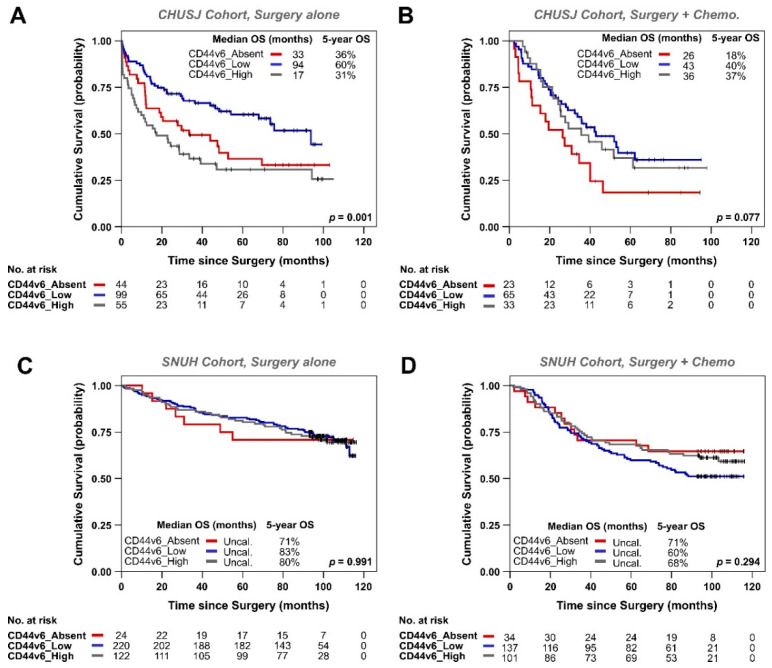
Kaplan–Meier estimates showing OS of GC patients according to CD44v6 status (absent, low, or high) in CHUSJ patients treated with surgery alone (**A**) or treated with surgery and conventional chemotherapy (**B**); SNUH patients treated with surgery alone (**C**) or with surgery plus conventional chemotherapy (**D**). Patients from all pTNM stages are included in this analysis. The tables below each graph indicate the number of patients still at risk in each group. Median OS and 5-year OS are shown for each patient subgroup. *p*-values for pairwise comparisons by the Log-Rank (Mantel–Cox test) for Figure 2A,B are shown in Appendix A, respectively.

**Figure 3 biomedicines-09-01249-f003:**
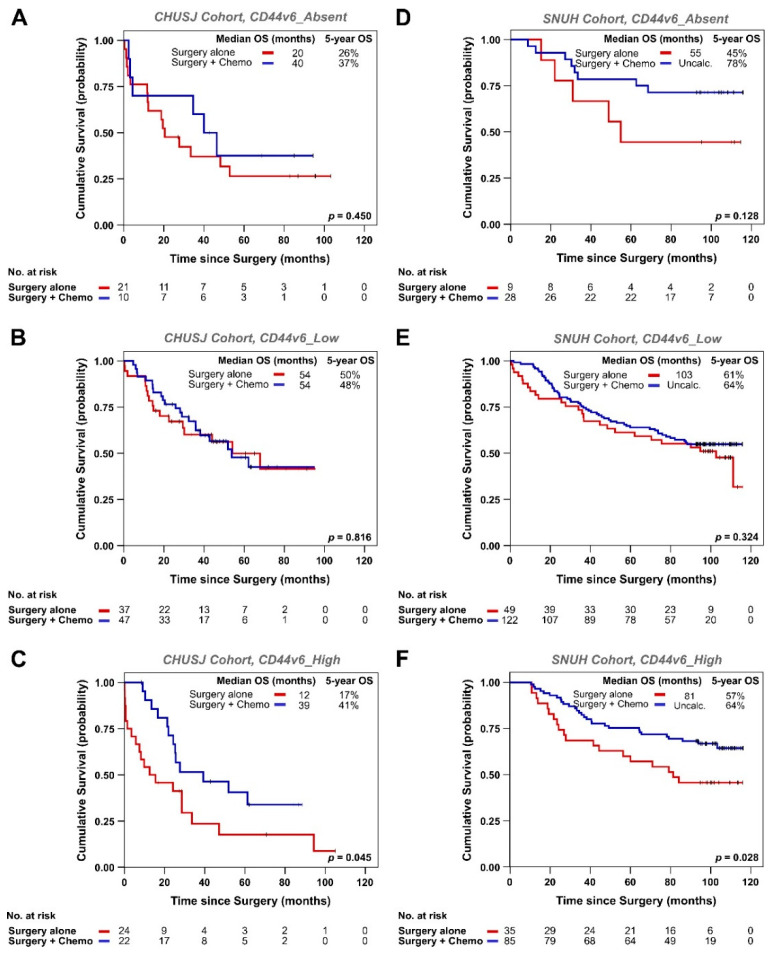
Kaplan–Meier estimates showing OS of GC patients treated with surgery alone (red) or surgery plus conventional chemotherapy (blue), according to CD44v6. OS of CHUSJ patients with CD44v6_Absent (**A**), CD44v6_Low (**B**), and CD44v6_High (**C**) tumors; OS of SNUH patients with CD44v6_Absent (**D**), CD44v6_Low (**E**), and CD44v6_High (**F**) tumors. The tables below each graph indicate the number of patients still at risk in each group. Only pTNM stage II and III patients are included in this analysis since >95% of stage I patients herein analyzed remained chemotherapy untreated and would bias this analysis, and stage IV patients are treated with palliative chemotherapy without curative intent. Median OS and 5-year OS are shown for each patient subgroup.

**Figure 4 biomedicines-09-01249-f004:**
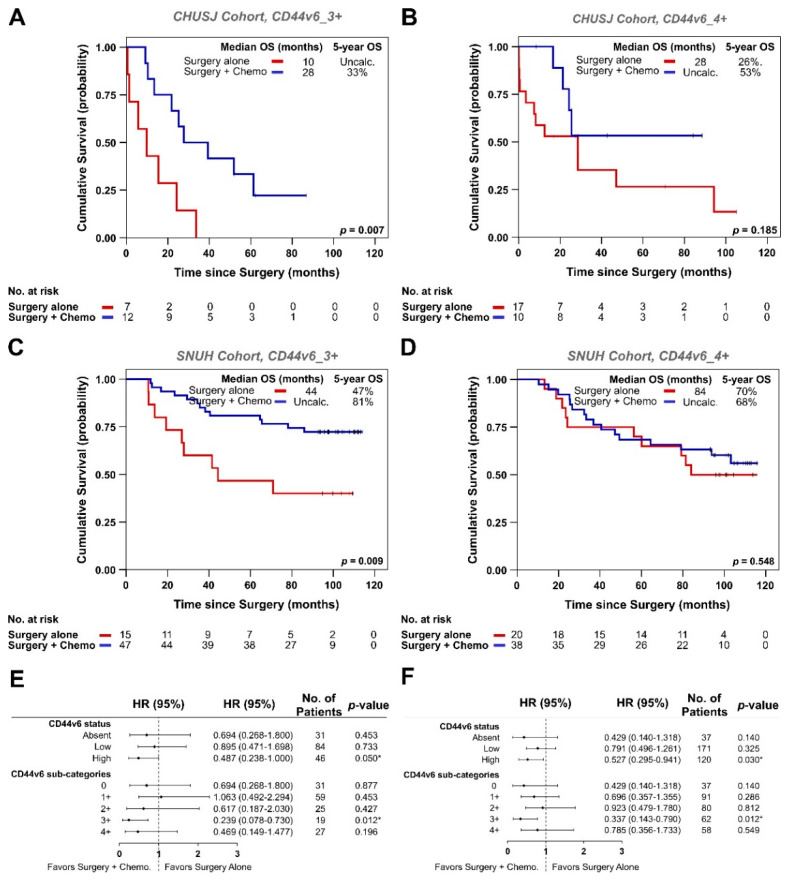
Kaplan–Meier estimates showing OS of GC patients treated with surgery alone (red) or with surgery plus conventional chemotherapy (blue). OS of CHUSJ patients harboring CD44v6_3+ (**A**) or CD44v6_4+ tumors (**B**); OS of SNUH patients harboring CD44v6_3+ (**C**) or CD44v6_4+ tumors (**D**). Only pTNM stage II and III patients are included. The tables below each graph indicate the number of patients still at risk in each group. Median OS and 5-year OS are shown for each patient subgroup; Forest plots showing the benefit of treating CHUSJ (**E**) or SNUH (**F**) GC patients with surgery plus chemotherapy according to CD44v6 classification. * highlights statistically significant differences.

**Table 1 biomedicines-09-01249-t001:** Clinicopathological associations with extent of CD44v6 expression in gastric tumors from the CHUSJ cohort.

Variables	Total No. Patients *n* = 326	CD44v6_0 *n* = 68/326 (21%)	CD44v6_1+ *n* = 101/326 (31%)	CD44v6_2+ *n* = 68/326 (21%)	CD44v6_3+ *n* = 46/326 (14%)	CD44v6_4+ *n* = 43/326 (13%)	*p*-Value
**Age (years)**							>0.05
Mean	67.7	68.0	67.5	66.2	69.9	69.9	
SD	11.8	12.3	10.5	13.5	11.9	11.9	
**Gender**							**0.005**
Male	185 (56.7%)	42 (61.8%)	65 (64.4%)	35 (51.5%)	29 (63.0%)	14 (32.6%)	
Female	141 (43.3%)	26 (38.2%)	36 (35.6%)	33 (48.5%)	17 (37.0%)	29 (67.4%)	
M:F ratio	1.3:1	1.6:1	1.8:1	1.1:1	1.7:1	0.5:1	
**Laurén classification**							>0.05
Intestinal	163 (50.0%)	31 (45.6%)	56 (55.4%)	35 (51.5%)	24 (52.2%)	17 (39.5%)	
Diffuse	44 (13.5%)	12 (17.6%)	11 (10.9%)	9 (13.2%)	5 (10.9%)	7 (16.3%)	
Mixed	84 (25.8%)	18 (26.5%)	18 (17.8%)	18 (26.5%)	15 (32.6%)	15 (34.9%)	
Indeterminate	35 (10.7%)	7 (10.3%)	16 (15.8%)	6 (8.8%)	2 (4.3%)	4 (9.3%)	
**Growth pattern**							>0.05
Expansive	60 (18.4%)	13 (19.1%)	19 (18.8%)	12 (17.6%)	10 (21.7%)	6 (14.0%)	
Infiltrative	252 (77.3%)	51 (75.0%)	77 (76.2%)	53 (77.9%)	35 (76.1%)	36 (83.7%)	
Unclassified	14 (4.3%)	4 (5.9%)	5 (5.0%)	3 (4.4%)	1 (2.2%)	1 (2.3%)	
**Lymphatic permeation *^1^**							>0.05
Absent	99 (30.5%)	21 (30.9%)	31 (31.0%)	26 (38.8%)	14 (30.4%)	7 (16.3%)	
Present	225 (69.2%)	47 (69.1%)	69 (69.0%)	41 (61.2%)	32 (69.6%)	36 (83.7%)	
**Perineural invasion *^1^**							**0.034**
Absent	166 (51.1%)	34 (50.0%)	59 (58.4%)	38 (55.9%)	22 (48.9%)	13 (30.2%)	
Present	159 (48.9%)	34 (50.0%)	42 (41.6%)	30 (44.1%)	23 (51.1%)	30 (69.8%)	
**Vascular invasion *^1^**							**0.005**
Absent	131 (40.6%)	25 (37.3%)	43 (42.6%)	38 (56.7%)	16 (35.6%)	9 (20.9%)	
Present	192 (59.4%)	42 (62.7%)	58 (57.4%)	29 (43.3%)	29 (64.4%)	34 (79.1%)	
**Surgical margins *^1^**							**0.007**
**R0**	290 (89.2%)	58 (85.3%)	98 (98.0%)	61 (89.7%)	37 (80.4%)	36 (83.7%)	
**R1/R2**	35 (10.8%)	10 (14.7%)	2 (2.0%)	7 (10.3%)	9 (19.6%)	7 (16.3%)	
**Depth of invasion (T)**							**0.010**
pT1	74 (22.7%)	14 (20.6%)	24 (23.8%)	24 (35.3%)	9 (19.6%)	3 (7.0%)	
pT2	41 (12.6%)	6 (8.8%)	14 (13.9%)	41 (60.3%)	10 (21.7%)	8 (18.6%)	
pT3–T4	211 (64.7%)	48 (70.6%)	63 (62.4%)	3 (4.4%)	27 (58.7%)	32 (74.4%)	
**Lymph node metastases (N) *^1^**							>0.05
Absent (pN0)	126 (38.8%)	27 (39.7%)	39 (39.0%)	29 (42.6%)	17 (37.0%)	14 (32.6%)	
Present (pN+)	199 (61.2%)	41 (60.3%)	61 (61.0%)	39 (57.4%)	29 (63.0%)	29 (67.4%)	
**Distant metastases (M)**							>0.05
Absent	257 (78.8%)	48 (70.6%)	89 (88.1%)	51 (75.0%)	35 (76.1%)	34 (79.1%)	
Present	69 (21.2%)	20 (29.4%)	12 (11.9%)	17 (25.0%)	11 (23.9%)	9 (20.9%)	
**TNM Staging**							>0.05
I	95 (29.1%)	17 (25.0%)	30 (29.7%)	25 (36.8%)	16 (34.8%)	7 (16.3%)	
II	74 (22.7%)	16 (23.5%)	27 (26.7%)	11 (16.2%)	7 (15.2%)	13 (30.2%)	
III	88 (27.0%)	15 (22.1%)	32 (31.7%)	15 (22.1%)	12 (26.1%)	14 (32.6%)	
IV	69 (21.2%)	20 (29.4%)	12 (11.9%)	17 (25.0%)	11 (23.9%)	9 (20.9%)	

*^1^ Data not available for < than 5 cases. *p*-values in bold highlight statistically significant results.

**Table 2 biomedicines-09-01249-t002:** Clinicopathological associations with extent of CD44v6 expression in gastric tumors from the SNUH cohort.

Variables	Total No. Patients *n* = 638	CD44v6_0 *n* = 58/638 (9%)	CD44v6_1+ *n* = 163/638 (26%)	CD44v6_2+ *n* = 194/638 (30%)	CD44v6_3+ *n* = 136/638 (21%)	CD44v6_4+ *n* = 87/638 (14%)	*p*-Value
**Age (years)**							>0.05
Mean	60.8	59.5	60.8	60.2	60.7	63.3	
SD	12.3	12.8	12.5	11.9	12.1	12.8	
**Gender**							**0.009**
Male	420 (65.8%)	39 (67.2%)	120 (73.6%)	131 (67.5%)	73 (53.7%)	57 (65.5%)	
Female	218 (34.2%)	19 (32.8%)	43 (26.4%)	63 (32.5%)	63 (46.3%)	30 (34.5%)	
M:F ratio	1.9:1	2.1:1	2.8:1	2.1:1	1.2:1	1.9:1	
**Laurén classification *^1^**							>0.05
Intestinal	299 (47.3%)	26 (44.8%)	67 (41.9%)	102 (52.8%)	64 (47.4%)	40 (46.5%)	
Diffuse	239 (37.8%)	29 (50.0%)	68 (42.5%)	58 (30.1%)	51 (37.8%)	33 (32.5%)	
Mixed	88 (13.9%)	2 (3.4%)	22 (13.8%)	33 (17.1%)	18 (13.3%)	13 (15.1%)	
Indeterminate *^2^	6 (0.9%)	1 (1.7%)	3 (1.9%)	0 (0.0%)	2 (1.5%)	0 (0.0%)	
**Growth pattern**							>0.05
Expansive	65 (10.2%)	6 (10.3%)	17 (10.4%)	20 (10.3%)	13 (9.6%)	9 (10.3%)	
Infiltrative	573 (89.8%)	52 (89.7%)	146 (89.6%)	174 (89.7%)	123 (90.4%)	78 (89.7%)	
**Lymphatic permeation**							**0.001**
Absent	309 (48.4%)	28 (48.3%)	82 (50.3%)	99 (51.0%)	76 (55.9%)	24 (27.6%)	
Present	329 (51.6%)	30 (51.7%)	81 (49.7%)	95 (49.0%)	60 (44.1%)	63 (72.4%)	
**Perineural invasion**							>0.05
Absent	405 (63.5%)	27 (46.6%)	103 (63.2%)	130 (67.0%)	88 (64.7%)	57 (65.5%)	
Present	233 (36.5%)	31 (53.4%)	60 (36.8%)	64 (33.0%)	48 (35.3%)	30 (34.5%)	
**Vascular invasion**							>0.05
Absent	536 (84.0%)	48 (82.8%)	133 (81.6%)	168 (86.6%)	114 (83.8%)	73 (83.9%)	
Present	102 (16.0%)	10 (17.2%)	30 (18.4%)	26 (13.4%)	22 (16.2%)	14 (16.1%)	
**Surgical margins *^3^**							
R0	638 (100%)	58 (100%)	163 (100%)	194 (100%)	136 (100%)	87 (100%)	
R1/R2	0 (0.0%)	0 (0.0%)	0 (0.0%)	0 (0.0%)	0 (0.0%)	0 (0.0%)	
**Depth of invasion (T)**							**0.0001**
pT1	212 (33.2%)	11 (19.0%)	53 (32.5%)	83 (42.8%)	45 (33.1%)	20 (23.0%)	
pT2	118 (18.5%)	7 (12.1%)	23 (14.1%)	39 (20.1%)	33 (24.3%)	16 (18.4%)	
pT3–T4	308 (48.3%)	40 (69.0%)	87 (53.4%)	72 (37.1%)	58 (42.6%)	51 (58.6%)	
**Lymph node metastases (N)**							**0.008**
Absent (pN0)	324 (50.8%)	25 (43.1%)	79 (48.5%)	113 (58.2%)	75 (55.1%)	32 (36.8%)	
Present (pN+)	314 (49.2%)	33 (56.9%)	84 (51.5%)	81 (41.8%)	61 (44.9%)	55 (63.2%)	
**Distant metastases(M)**							>0.05
Absent	596 (93.4%)	52 (89.7%)	153 (93.9%)	186 (95.9%)	125 (91.9%)	80 (92.0%)	
Present	42 (6.6%)	6 (10.3%)	10 (6.1%)	8 (4.1%)	11 (8.1%)	7 (8.0%)	
**TNM Staging**							**0.001**
I	269(42.2%)	15 (25.9%)	62 (38.0%)	106 (54.6%)	63 (46.3%)	23 (26.4%)	
II	158 (24.8%)	16 (27.6%)	45 (27.6%)	36 (18.6%)	34 (25.0%)	27 (31.0%)	
III	170 (26.6%)	21 (36.2%)	46 (28.2%)	44 (22.7%)	28 (20.6%)	31 (35.6%)	
IV	41 (6.4%)	6 (10.3%)	10 (6.1%)	8 (4.1%)	11 (8.1%)	6 (6.9%)	

*^1^ Data not available for less than 7 cases. *^2^ “Indeterminate” cases were not included in the statistical analysis due to the low number of cases, which would invalidate the chi-square test result. *^3^ All cases were classified as R0 for the surgical margins, so no statistical test was performed. *p*-values in bold highlight statistically significant results.

## Data Availability

The data presented in this study are available in a Appendix A.

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
