# Peer review of "CD44v6 High Membranous Expression Is a Predictive Marker of Therapy Response in Gastric Cancer Patients"

_biomedicines, 2021, doi:10.3390/biomedicines9091249_

Round 1

Reviewer 1 Report

In this paper, the authors tackle the challenging issue of the clinical relevance of the search of innovative predictive and prognostic biomarkers for gastric cancer (GC) to improve patients’ stratification and treatment.

Using an analytical framework, the authors report the results of their translational knowledge in this field. The authors’ work is aimed to investigate the role of CD44v6 membranous tumor expression in GC.

The Introduction section provides a good, generalized background of the topic that quickly gives the reader a couple of useful key information to understand the setting and the aims of the paper. 

The discussion part of the paper is interesting and well written. 

The overall level of the paper is pretty good and innovative: the hypothesis of a new useful biomarker for GC patients is intriguing, the main messages are clearly stated and some important conclusions about the potential value of CD44v6 as a predictive biomarker have been confirmed by the experimental plan detailed in the text and the robust statistical analysis.

Author Response

Thank you for your review and kind comments on the manuscript which we appreciate.

Reviewer 2 Report

Very complete and interesting paper, however I have the following comments that I hope you take into consideration:

  • Line 86: when you mention an increase in CD44, are you referring to both CD44s and CD44v?
  • It would be interesting if in the introduction they explained why they specifically focus on the CD44v6 isoform or what characterizes this isoform.
  • Line 135 and line 226: I suggest modifying the titles of the sections because more than a title it seems a conclusion
  • Figure 2, 3 and 4: I am not clear what the tables below each graph mean. I suggest that its meaning be explained.
  • I find it striking that the OS data in figure 2 and 3 are different, this is clarified in the legend of figure 3 “Only pTNM stage II and III patients are included in this analysis, since> 95% of stage I patients herein analyzed remained chemotherapy untreated and would bias this analysis, and stage IV patients are treated with palliative chemotherapy without curative intent ”but I suggest giving this explanation in the text (in the paragraph starting on line 251) to improve the understanding of what It is explained and why the data that is spoken does not coincide with the ones shown just above.

Reviewer 3 Report

Very complete and interesting paper, however I have the following comments that I hope you take into consideration:

  • Line 86: when you mention an increase in CD44, are you referring to both CD44s and CD44v?
  • It would be interesting if in the introduction they explained why they specifically focus on the CD44v6 isoform or what characterizes this isoform.
  • Line 135 and line 226: I suggest modifying the titles of the sections because more than a title it seems a conclusion
  • Figure 2, 3 and 4: I am not clear what the tables below each graph mean. I suggest that its meaning be explained.
  • I find it striking that the OS data in figure 2 and 3 are different, this is clarified in the legend of figure 3 “Only pTNM stage II and III patients are included in this analysis, since> 95% of stage I patients herein analyzed remained chemotherapy untreated and would bias this analysis, and stage IV patients are treated with palliative chemotherapy without curative intent ”but I suggest giving this explanation in the text (in the paragraph starting on line 251) to improve the understanding of what It is explained and why the data that is spoken does not coincide with the ones shown just above.

Author Response

Reviewer 3 is the same as reviewer 2.

Please see the previous attachment.